# Study of the Impact of Rural Land Transfer on the Status of Women in Rural Households

**Mingyong Hong, Donglai Zhou * and Lei Lou**

School of Economics, Guizhou University, Guiyang 550025, China; hongmingyong@163.com (M.H.); gs.llou21@gzu.edu.cn (L.L.)
* Correspondence: xd_dlzhou@163.com

**Abstract:** While the status of rural women in the family has undergone changes, rural land transfer has brought about transformations in both rural production and daily life. This paper adopts the perspective of rural land transfer, follows the research track of Marx and Engels's theory of women, and based on the theoretical research of the changes in the status of modern women in the family, constructs a framework for analyzing the status of women in rural families. Drawing on the data from the 2014 China Family Panel Studies (CFPS2014), this article utilizes OLS (Ordinary Least Square) and ordered logit models to explore the impact of rural land transfer on the status of women in rural households. The study reveals the following findings: Initially, rural land transfer-out improves women's household decision-making power and enhances the status of women in rural households. The reliability of these results is further confirmed through robustness tests and endogeneity discussions. Secondly, the heterogeneity analysis indicates that the transfer of agricultural land promotes the status of women in rural households in nonmajor grain-producing areas more than women in major grain-producing areas. The reason is that women in major grain-producing areas lack off-farm employment opportunities compared with women in non-major grain-producing areas and the main grain producing areas may have a strong patriarchal cultural atmosphere. Thirdly, the analysis of mechanisms indicates that rural land transfer-out improves the status of women in rural households by augmenting their independent income. Conversely, rural land transfer-in increases women's private labor and decreases their independent income without promoting their family status. The study sheds light on rural women's empowerment, the improvement of intra-household bargaining power, and the comprehensive development of rural women. The conclusion of this paper provides a new understanding and some recommendations for us to explore the change of rural women's status in the family.

**Keywords:** rural land transfer; female empowerment; women's family status; CFPS2014

## 1. Introduction

The development of women's careers and the protection of their rights and interests have consistently been global focal points, and China is no exception. The Twentieth National Congress of the Communist Party of China (CPC) explicitly restated its commitment to "adhere to the basic State policy of equality between men and women and safeguard the lawful rights and interests of women and children", emphasizing the crucial role of acknowledging that "women are capable of holding up half of the sky". According to the 2020 Survey on the Social Status of Chinese Women, the proportion of women participating in decision-making in major family affairs such as "investment/loans" and "buying/building a house" has significantly risen to 89.5% and 90.0%, respectively, reflecting increases of 14.8% and 15.6% compared to 2010. However, in rural areas, the enduring gender role concept and division of labor, where "men dominate the outside, females dominate the inside", persist. Women continue to shoulder the primary responsibilities for housework and children's education [1]. This situation, discrimination in the labor market [2], contributes

to the lower status of women in rural households. Nonetheless, rural women have long constituted the backbone of the rural labor force [3], and any shift in the status of women in rural households significantly influences the realization of the "rural revitalization" strategy in China.

Research on the factors influencing changes in the status of women in rural households has predominantly concentrated on the micro-levels, such as income [4–6], subjective identity [7], various types of labor participation, and so forth [8,9], but has often neglected the impact of changes in the rural institutional environment. The continuous promotion of the rural land transfer policy has effectively increased the transfer ratio, bringing significant changes to rural production and life [10]. Rural land transfer has spurred the migration of the rural labor force to urban areas [11]. This not only leads to a substantial increase in the proportion of non-agricultural employment but also facilitates the optimal allocation of rural labor resources [12,13]. Such changes may have a certain impact on the original decision-making power within families.

Does the transfer of agricultural land shift the locus of household decision-making in favor of women, leading to an improvement in the status of women in rural households? What is the impact of both overall farmland transfer and specific types of farmland transfer on the status of women in rural households? Furthermore, what underlying mechanisms drive these effects? To address these questions, this paper utilizes nationally representative data from the 2014 China Family Panel Studies (CFPS2014), comprising 7735 participants, which includes nearly all rural female respondents. The aim is to explore both theoretically and empirically the effects of farmland transfer on the status of women in rural households and to uncover the mechanisms at play. Additionally, this research seeks to examine the heterogeneity of the effects of farmland transfer on the status of women in rural households, considering variations between major grain-producing areas and non-major grain-producing areas. These inquiries form the core focus of this paper.

## 2. Literature Review

Women's roles, characterized as "a set of social identities and behavioral norms", encompass their positions and behaviors within society. In the context of the family, the status of women pertains to the esteem they hold within the family structure and their capacity to own and manage the family's resources. More specifically, the status of women in the family is reflected in their entitlement to make decisions regarding significant family matters [14,15]. This includes rights such as managing their income, disposing of their earnings, making consumption decisions, and participating in decisions related to the education of their children, among others [8,14].

Current explanations for changes in the status of women in households primarily center around the resource-determination theory perspective. According to this theory, the spouse possessing more resources holds a higher family status within the household [16]. Earlier studies predominantly focused on tangible resources, such as income and natural resources. In terms of income, women's engagement in non-agricultural labor is associated with higher earnings. Those participating in non-agricultural labor often experience increased equality and subjective well-being in family life [17], influencing original family power dynamics and, consequently, decision-making power in family affairs [18]. Regarding natural resources, land, a crucial asset in rural areas, has been a focal point. Research by Hou and Omondi reveals that women's ownership of land significantly enhances their social status [19,20]. Meanwhile, Rao, in a study conducted in water-scarce South Asia, argues that women's control over water resources directly determines their household status [21]. Subsequent studies have broadened the scope to include both tangible and intangible resources [22–24]. Intangible resources, such as education and self-identity, have been successively incorporated into the resource category influencing family status [25]. Education, as an intangible resource, empowers the more educated spouse to have a more prominent role in managing family affairs [26]. Self-identity, considered a subjective feeling, plays a crucial role in consolidating the status of women in households. Wang and Li

found that, with the increasingly stringent selection conditions for party members, party membership enhances women's self-identity, contributing to the awakening of their sense of rights and the improvement of their status within households [27].

Cultural norms theory proposes that patriarchal cultural norms play a crucial role in shaping the distribution of power within the family [28]. Within the framework of a patriarchal culture, individual characteristics of women and their spouses, such as age, education, and personality, can significantly influence the status of women in families [29,30]. In developing countries, the number of births, particularly the number of boys, is identified as having a decisive impact on the status of women in families [31]. In this context, women act in strict accordance with gender norms, which disempower them [32]. Women's work outside the home is stigmatized and women are allowed to work only in low-paying agricultural jobs and are dependent on their husbands, weakening the status of the family [33]. The patriarchal culture has also spawned a phenomenon in traditional Chinese society, in which the mother is valued by the son. Women need to rely on their children to earn their place in the family [34]. The strong patriarchal culture in East Asia is an important reason for the low status of rural women in the family.

In the rice district, women exhibit a comparative advantage in intensive rice farming compared to men, thereby wielding strong intra-family bargaining power. Conversely, in the wheat district, the labor market preference for men with physical strength results in the marginalization of the female labor force and a corresponding weakening of the status of women within families [9]. From the perspective of cash crop production, Wu et al. observed that women involved in tea picking enjoyed higher incomes, experienced fewer spousal quarrels, and held a higher status within their families compared to women not engaged in tea picking. Furthermore, women of higher status contributed to creating a positive atmosphere for their children to grow up in the family [10].

In summary, whether viewed through the lens of resource determinism, cultural norms, or the analysis of women's family status under various types of labor force participation, all perspectives underscore the crucial role of women's ability to access resources in shaping their family status within households. While existing studies offer a solid theoretical foundation for this paper, there remains room for more in-depth research. Therefore, the study will make efforts in the following aspects: Firstly, the relevant literature has predominantly explored changes in women's status in rural households from the vantage point of resource determinism, often regarding rural women's resource endowment as scarce. This study aims to expand this perspective by examining the status of women in rural families from the angle of rural land resources. By subdividing farmland transfer behavior into transfer-out and transfer-in, the study seeks to accurately identify the relationship between these actions and women's status in rural households. Robustness tests will be conducted to enhance the realism and reliability of the estimation results. Secondly, while most previous studies have constructed theoretical frameworks based on Western economic theories to explain changes in women's family status, this article takes a different approach. It initiates the research path from Marx and Engels' women's theories, establishing a novel analytical framework of "transfer of farmland-resource grabbing ability-family status". This fresh perspective provides new insights into women's empowerment and enriches the understanding of rural women's status. Thirdly, existing studies have given limited attention to the impact of agricultural land transfer on women's social status [19,20]. However, women's social status is not equivalent to the status within households. Few studies directly explore the impact of agricultural land transfer on the status of women in families and the internal transmission mechanism. Furthermore, conducting a heterogeneous analysis on this basis is not only conducive to advancing China's women's causes but will also aid in better fulfilling the role of "women can hold up half the sky" in the construction of rural revitalization.

## 3. Theoretical Analysis

The status of women in families has evolved through various stages of human history. Engels, in "The Origins of the Family, Private Ownership, and the State", conducted a

systematic analysis of these changes [35]. In primitive society, the status of women in families exhibited a pattern of "equal—higher—relatively equal". During the primitive communist period, where survival depended on hunting, men and women hunted together, distributing resources evenly and resulting in equal status between men and women in families. Transitioning to the matrilineal clan period, primitive agriculture emerged, and women's gathering activities became a stable source of food for the family. Their advantage in gathering food elevated their status within the family. In the patrilineal clan period, where livelihoods relied on animal husbandry, men dominated in productive life due to their physical strength, positioning them in the first or second place within the family. During this period, the status of men and women became relatively equal. It is evident that in both patriarchal and matrilineal societies, the distribution of power is fundamentally determined by resource-grabbing ability, whether possessed by men or women.

Drawing on Engels' resource theory, Blood and Wolfe developed a relative resource theory to elucidate power and status dynamics within the family [16]. According to this framework, intra-family power is contingent upon the relative resources possessed by each spouse. If one partner holds more resources in terms of education, occupation, income, and social participation, their status within the family is elevated [36]. For example, Bertrand et al. found that the relative income of a couple is an important indicator of family bargaining power and determines their respective status in the family [37]. The strength of each spouse's resource-capturing ability determines their respective family status. The economic income earned by women represents the resource grabbing capacity we mentioned earlier. Building upon Blood and Wolfe's theory, Heer introduced the resource exchange theory [22]. Safilios-Rothschild proposed the relative love and need theory [23], while Rodman formulated a resource theory within the cultural context [24]. These theories endorse income-centered resource determinism and broaden the scope of resources influencing women's family status to include education, kinship, the number of children born, and physical attributes. As society evolves, more quantifiable and comparable resources are supplanting traditional natural resources as determinants of the status of both men and women within families [38]. In the research on the status of Indian women, Deininger found that women's right to inherit property could improve their autonomy and status [39]. Dong found in his research on house ownership in China that house ownership would change women's bargaining power [40]. The inferiority of women in the ownership of family property has reduced women's family status [41]. Han found in his research on Chinese women that mobile money can improve women's control over family assets and enhance their status in family decision-making [42]. In the following, we summarize the studies of scholars in different periods on the factors affecting women's family status in a tabular form, as shown in Table 1. As can be seen from Table 1, although the factors affecting women's status in the family change over time, they are not dependent on income.

**Table 1.** Research on women's family status in different periods.

| Periods | Researchers | Factors Affecting Women's Status in the Family |
|---|---|---|
| The 19th century | Engels and Max [35] | Independent income |
| The 20th century | Blood and Wolfe (1960) [16] Heer (1963) [22]; Safilios-Rothschild (1967) [23]; Rodman (1973) [24] | Education; Kinship; number of children; appearance |
| The 21st century | Deininger (2013) [39] Fortin (2015) [43] Bertrand et al. (2015) [37] Duman (2021) [33] Dong (2022) [40] Han (2023) [42] | Right of inheritance Quality of employment and income level Relative income The patriarchal culture House ownership Mobile money and online banking |

As Aizer said, only when women are economically empowered can their bargaining power in the family be improved [44]. In the current era, rural areas are transitioning from self-sufficiency to deep integration with urban areas. The independent economic income acquired by women reflects their resource-grabbing ability, with income gradually becoming the core factor in determining women's status within families [15]; money earned by women empowers women. Men gained authority in the home by their ability to be breadwinners [45]. Consequently, women can only experience an improvement in their family status by becoming independent income earners. Based on the above theories, we constructed a theoretical analysis framework of the impact of rural land transfer on women's status in the family—"Rural land transfer—resource grabbing ability—family status". The following is a specific analysis and Figure 1 is the mechanism diagram of the study.

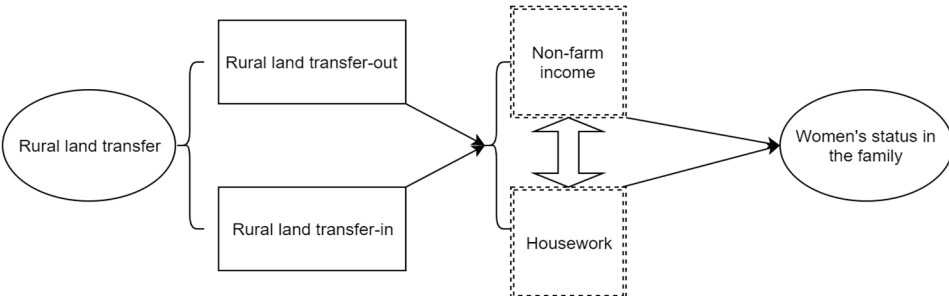

**Figure 1.** The mechanism diagram of the influence of farmland transfer on women's family status.

Rural land transfer significantly impacts the status of women in households, primarily through changes in income (with a specific focus on non-farm income) and the reorganization of household responsibilities. On one hand, the process of rural land transfer-out emancipates women from traditional agricultural roles, paving the way for increased participation in off-farm employment. This transition allows women to command significantly higher wages than what would have been possible through their previous engagement in agricultural production. As previously highlighted, the level of resource extraction capacity plays a pivotal role in determining family status, with women's influence manifested in their decision-making authority over crucial family matters [14]. The elevation of women's independent economic income enhances their resource-capturing capabilities, promoting rights associated with income management, control, and consumption decisions [8]. Consequently, there is an overall improvement in household status. At this juncture, women often advocate for the equitable redistribution of housework between genders or the communalization of housework. This redistribution not only results in increased satisfaction with labor division within the household but also translates into higher economic income for women, rooted in their adept management of household affairs and contributions to services [25]. This, in turn, further enhances the status of women within the household. Moreover, the redistribution of housework creates opportunities for women to make additional investments in their human capital. The self-reinforcing trend of women's participation in the non-agricultural labor market continually refines their skills [46]. As women become more proficient, their incomes increase, contributing to the sustained long-term improvement of their status in families.

On the other hand, not only did the time dedicated to women's housework remain largely unchanged after rural land transfer-in, but the rise in women's private labor within the household [47] and the reduction in independent income available to women also hindered the improvement of the status in the family. This is because, despite contributing to family income, women's private labor remains within the realm of private labor and services. These efforts are often taken for granted by society and the family, and women are not remunerated for their contributions [48]. The fruits of women's private labor are uniformly distributed within the family, while the opportunity cost of such labor is borne by

the women themselves. Consequently, women engaged in agricultural land transfer-in find themselves less capable of resource capture and receive less independent economic income compared to those involved in transfer-out. This disparity results in minimal changes in the status in the family. Differences in the quality of employment and income levels have led to women lagging behind [43]. The consistent allocation of women's time to housework also implies an underinvestment in their human capital, hindering improved resource capture and impeding long-term advancements in household status. Rural land transfer-in does not enhance women's resource capture and, consequently, does not improve their status within households.

This leads to the hypothesis of this paper:

**H1:** *The status of women in rural households is enhanced by rural land transfer-out, whereas rural land transfer-in does not have the same effect;*

**H2:** *Rural land transfer-out enhances the status of women in rural households by increasing their non-farm income and reducing the time spent on housework;*

**H3:** *Rural land transfer-in reduces women's income and increases the time spent on housework, which is detrimental to the improvement of the status of women in rural households.*

This paper will test H1 in Section 5.1 Baseline regression. H2 and H3 will be tested in Section 6.

## 4. Data, Variables, and Research Design

### 4.1. Data Sources

This paper examines the impact of rural land transfer on the status of women in rural families, using micro-survey data from the 2014 China Family Panel Studies (CFPS), a nationwide tracking survey program conducted every two years by Peking University's Social Science Research. The survey consists of three levels of questionnaires: individual, household, and community (or village), covering information on China's economy, society, education, and health. The sample of the survey covers 25 provinces/municipalities/autonomous regions in China, excluding Hong Kong, Macau, Taiwan, Hainan, Xinjiang, Tibet, Qinghai, Ningxia, and Inner Mongolia. Although the CFPS is tracking data, the content of the questionnaire focuses on different aspects of each issue. The 2014 CFPS data are employed for empirical research for the following reasons: First, existing national microdata surveys in China have given less attention to women's status in rural households. Fortunately, the 2014 CFPS data survey helped fill this gap. Second, the 2014 CFPS allows for an examination of intra-household bargaining power, including household expenditures, savings, financial investments, children's education, and the purchase of housing and high-end durable goods. These comprehensive data enable a more nuanced measurement of the status of women in rural households. Finally, previous studies on the status of women in rural households have used relevant questions from this dataset, such as Li et al. [7] and Wang et al. [27], establishing feasibility and providing an academic foundation for this paper's research. The 2014 CFPS data contains a total of more than 37,000 nationally representative samples and the data processing process was as follows:

(1) the individual, household, and community pools in the CFPS database were matched and merged;

(2) the urban samples and rural unmarried samples were deleted, yielding a rural sample of 18,349;

(3) males and repeated samples were deleted from the rural samples, resulting in a final sample of 7735 female respondents from across the country. The data used in this paper are all rural females in the database.

It is crucial to note that the CFPS database's multistage stratified PPS sampling design ensures a representative sample of 95% of the Chinese population, maintaining an even

gender distribution and a reasonable age and income structure. This scientifically rigorous and broadly representative sample mitigates selective bias and provides insights into the real situation in China.

*4.2. Statistical Description*

Based on the research objectives of our study and the results of existing studies, the following variables are proposed.

### 4.2.1. Dependent Variables

To study the change in the status of women in rural households, a reasonable measure of women's status is crucial. When considering the status of women in rural households, it is important to consider women's ability to occupy the disposal of resources and decision-making power in family affairs [15], which is compared with men [14,49]. The measurement of the status of women in rural households has a horizontal and vertical relationship. The former refers to the husband and wife [50] while the latter refers to the comparison between the parent generation (mother-in-law and daughter-in-law). In the author's view, given the current imbalance in the sex ratio of marriageable men and women and the difficulty of getting married in rural areas, the status of mothers-in-law and daughters-in-law should not be the focus of research on the status of women in rural families compared with the status of husband and wife. Furthermore, some scholars believe that the measurement of the status of women in rural households does not depend on the relative power of the wife. The advancement of women's status is not at the expense of men's status, but rather the pursuit of the relative equality of the status of husband and wife, aiming to the establishment of an equal and harmonious partnership [23]. Therefore, it is more appropriate to measure the status of women in rural households in terms of absolute rather than relative power. This paper draws on the research method of Li et al. [7]. We try to use the data from the 2014 China Family Panel Survey (CFPS) to study the impact of rural land transfer on the status of women in rural households, using the questionnaire's five major aspects of family affairs "Who is in charge? " to measure women's status in the family. These five aspects include the distribution of household expenditures, savings, and insurance investments, buying a house, parenting, and purchasing high-priced consumer goods. It means the decision-making power in major family matters represents women's status in the family. If the wife is in charge of this matter, the value is 1, and if the husband is in charge of it, the value is 0. After constructing five dummy variables for the above five questions, the scores will be summed up to form a comprehensive variable reflecting the status of women in rural families, "*Status*1". At the same time, this paper adopts the principal component analysis method to extract a common factor reflecting the status of women in rural households, named "*Status*2" (The KMO value of 0.892 indicates that the above five factors are suitable for factor analysis, and a common factor that explains 87% of the original variance fluctuation was obtained by Kaiser's standardized orthogonal rotation method).

### 4.2.2. Focus Variables

Referring to the research methods of Qian and Hong [51], Zhou et al. [52], Li et al. [53], and Hong and Lou [54], our study considers rural land transfer-out and rural land transfer-in as the core independent variables, which is mainly based on the two questions of "whether the land is leased to others" and "whether the land is rented from others" in the CFPS questionnaire of 2014. "Whether the land is leased to others" was selected as the measure of agricultural land transfer out, called "*Landout*", with a value of 1 for yes and 0 for no. Meanwhile, " Whether the land is rented from others " was also selected to measure rural land transfer-in, called "*Landin*".

4.2.3. Control Variables

To reduce estimation bias due to omitted variables, this paper draws on previous studies to control for factors that affect the status of women in rural households, including individual-level, household-level [7,21], and village-level variables [9]. Individual characteristic variables included age, education, health (unhealthy = 1; average = 2; relatively healthy = 3; very healthy = 4; very healthy = 5), and whether or not they owned property (yes = 1, no = 0). Household-level control variables include the spouse's age, spouse's education, number of children, and number of boys. It should not be overlooked that China is a large agricultural country with a long history, and the need for male labor in agricultural production has led to the prevalence of a culture of "more children, more happiness" in rural society. The birth of a male child results in a higher status for rural women rather than a girl. Therefore, this paper excludes the influence of rural culture on women's status by controlling for the number of boys women have. Village-level control variables include village per capita income and the proportion of women working outside the village. The definitions and descriptions of the main variables are presented in Table 2.

**Table 2.** Description of variables and descriptive statistics for all rural female samples.

| Variables | Definition | Mean | SD | Min | Max | Obs |
|---|---|---|---|---|---|---|
| *Status*1 | Takes values from 0 to 5 | 1.6804 | 2.1247 | 0 | 5 | 7735 |
| *Status*2 | factor score | 0 | 1 | −0.789 | 1.56 | 7735 |
| *Landout* | Whether land is leased (1 = yes; 0 = no) | 0.1119 | 0.3143 | 0 | 1 | 7735 |
| *Landin* | Whether land is rented (1 = yes; 0 = no) | 0.1707 | 0.3762 | 0 | 1 | 7735 |
| Age | Age of respondents | 41.5288 | 12.4019 | 20 | 63 | 7735 |
| Education | Years of education | 5.0649 | 4.54682 | 0 | 15 | 7735 |
| Estate | Whether you own a house (1 = yes; 0 = no) | 0.2723 | 0.4452 | 0 | 1 | 7735 |
| Health | Health status of the householder (from bad to good, 1–5) | 2.962 | 1.2972 | 1 | 5 | 7735 |
| Sage | Age of respondents' husband | 43.4047 | 12.8582 | 20 | 69 | 7735 |
| Seducation | Years of education of husband | 7.064 | 4.3753 | 0 | 18 | 7735 |
| Children | number of children | 2.0804 | 1.3468 | 0 | 10 | 7735 |
| Boys | number of boys | 1.1404 | 0.9373 | 0 | 6 | 7735 |
| Out percent | The proportion of the labor force working outside | 24.0764 | 18.7097 | 0 | 90 | 6975 |
| Average income | Per capita income in villages | 4872.04 | 4214.365 | 150 | 45,000 | 6975 |
| Housework | Time spent per housework | 2.262 | 1.9822 | 0 | 23 | 6676 |
| Non-farm income | Women's income | 16,771.29 | 24,228.92 | 0 | 220,000 | 6676 |

Note: Missing values for variables out percent, average income, housework, and non-farm income.

Table 2 presents descriptive statistics of the variables required in the regression equation. Descriptive statistics for the corresponding variables for the rural male and all rural samples are given in Tables 3 and 4. The dependent variable, *Status*1, takes values from 0 to 5 and has a mean value of 1.6804 in Table 2 while the value of *Status*1 is 3.0862 in Table 3, showing that rural women have a lower status in families. Specifically speaking, women participate in 1 to 2 significant family decisions on average. The mean values of focus variables, *Landout* and *Landin*, are 0.1119 and 0.1707, respectively, indicating that 11.19% and 17.07% of the rural female respondents in the sample have transferred farmland out and transferred in. This means that more than 80% of the rural female respondents have not carried out land transfers. The data show that despite the successive release of policy documents on land transfer, rural land transfer did not receive a good response from farmers in rural areas in 2013. The average age of the rural female respondents and their spouses is 41–43 years old in Table 1, with a standard deviation of 20, suggesting that the current population living in rural areas is predominantly middle-aged and old. In terms of

education, the mean value of years of education for women is 5.0649 in Table 1 while for the men is 7.6600 in Table 3, which simply means that the education level of the rural female respondents in the sample is mostly in elementary school and husbands in middle school. Husbands have a higher education level than their wives both in Tables 2 and 3. Compared with their husbands, women are at a relative disadvantage in terms of education. According to health, 2 to 5 in the questionnaire indicates good health. The mean value of women's health is 2.9613, which means that most of the rural female respondents are in good health. Regarding fertility in Tables 2 and 3, the mean value of the number of children is two, literally meaning that the rural female respondents have two children for their families on average. The maximum value is 10. It simply refers to the fact that the maximum number of children in the respondents' families is 10. Considering that the concept of "preferring sons over daughters" may still exist in rural areas and affect the status of women in the family, the statistics show that the average number of boys born to women in the sample is one, and the maximum number of boys in a single family is six. The mean value for property ownership is 0.2723, indicating that only 27.23% of the respondents own their property in our rural female samples while 56.237% of the respondents own property in rural male samples.

**Table 3.** Description of variables and descriptive statistics for all rural male samples.

| Variables | Definition | Mean | SD | Min | Max | Obs |
|---|---|---|---|---|---|---|
| *Status*1 | Takes values from 0 to 5 | 3.0862 | 2.241 | 0 | 5 | 10,614 |
| *Status*2 | factor score | 0.178 | 1.058 | −0.807 | 1.609 | 10,614 |
| *Landout* | Whether land is leased (1 = yes; 0 = no) | 0.1203 | 0.3253 | 0 | 1 | 10,614 |
| *Landin* | Whether land is rented (1 = yes; 0 = no) | 0.1586 | 0.3653 | 0 | 1 | 10,614 |
| Age | Age of respondents | 44.0887 | 17.2215 | 20 | 77 | 10,614 |
| Education | Years of education | 7.66 | 4.32 | 0 | 18 | 10,614 |
| Estate | Whether you own a house (1 = yes; 0 = no) | 0.5623 | 0.3227 | 0 | 1 | 10,614 |
| Health | Health status of the householder (from bad to good, 1–5) | 2.784 | 1.268 | 1 | 5 | 10,614 |
| Sage | Age of respondents' wife | 43.0628 | 15.3125 | 20 | 66 | 10,614 |
| Seducation | Years of education of wife | 5.8221 | 4.0988 | 0 | 18 | 10,614 |
| Children | number of children | 2.5803 | 1.0021 | 0 | 11 | 10,614 |
| Boys | number of boys | 1.2113 | 0.8372 | 0 | 7 | 10,614 |
| Out percent | The proportion of the labor force working outside | 37.0885 | 22.4729 | 0 | 90 | 10,614 |
| Average income | Per capita income in villages | 5112.167 | 4745.959 | 150 | 45,000 | 10,614 |
| Housework | Time spent per housework | 0.9503 | 1.864 | 0 | 19 | 10,614 |
| Non-farm income | Men's income | 23,600.9 | 28,630.37 | 0 | 326,800 | 10,614 |

Note: Missing values for variables out percent, average income, housework, and non-farm income.

### 4.2.4. Model Setting

To test the impact of land transfer on the status of women in rural households, this paper takes "*Status*1" and "*Status*2" as the dependent variables, meanwhile chooses "*Landout*" and "*Landin*" as the focus variables. At the same time, this paper controls as much as possible a series of other factors affecting the status of women in rural families and establishes the following model:

$$Status1_i = \beta_1 \times Landout_i + \gamma_1 \times X_1 + \delta_1 \times X_2 + \theta_1 \times X_3 + \varepsilon_1 \tag{1}$$

$$Status1_i = \beta_2 \times Landin_i + \gamma_2 \times X_1 + \delta_2 \times X_2 + \theta_2 \times X_3 + \varepsilon_2 \tag{2}$$

$$Status2_i = \alpha_1 + \beta_3 \times Landout_i + \gamma_3 \times X_1 + \delta_3 \times X_2 + \theta_3 \times X_3 + \varepsilon_3 \tag{3}$$

$$Status2_i = \alpha_2 + \beta_4 \times Landin_i + \gamma_4 \times X_1 + \delta_4 \times X_2 + \theta_4 \times X_3 + \varepsilon_4 \qquad (4)$$

**Table 4.** Description of variables and descriptive statistics for all rural samples.

| Variables | Definition | Mean | SD | Min | Max | Obs |
|---|---|---|---|---|---|---|
| *Status*1 | Takes values from 0 to 5 | 2.6522 | 1.8611 | 0 | 5 | 18,349 |
| *Status*2 | factor score | 0 | 0.998 | −0.807 | 1.609 | 18,349 |
| *Landout* | Whether land is leased (1 = yes; 0 = no) | 0.1185 | 0.3232 | 0 | 1 | 18,349 |
| *Landin* | Whether land is rented (1 = yes; 0 = no) | 0.1558 | 0.3627 | 0 | 1 | 18,349 |
| Age | Age of respondents | 44.3291 | 17.4615 | 20 | 77 | 18,349 |
| Education | Years of education | 6.8146 | 5.4552 | 0 | 18 | 18,349 |
| Estate | Whether you own a house (1 = yes; 0 = no) | 0.3782 | 0.263 | 0 | 1 | 18,349 |
| Health | Health status of the householder (from bad to good, 1–5) | 2.9247 | 1.2947 | 1 | 5 | 18,349 |
| Sage | Age of respondents' spouse | 44.0233 | 16.6148 | 20 | 77 | 18,349 |
| Seducation | Years of education of the spouse | 5.1322 | 4.2466 | 0 | 18 | 18,349 |
| Children | number of children | 2.2033 | 1.0231 | 0 | 11 | 18,349 |
| Boys | number of boys | 1.1611 | 1.1073 | 0 | 7 | 18,349 |
| Out percent | The proportion of the labor force working outside | 37.2185 | 22.5433 | 0 | 90 | 11,374 |
| Average income | Per capita income in villages | 5145.708 | 4716.483 | 150 | 45,000 | 11,374 |
| Housework | Time spent per housework | 1.8683 | 2.0526 | 0 | 23 | 11,673 |
| Non-farm income | Respondent's income | 17,042.16 | 15,805.51 | 0 | 326,800 | 11,673 |

Note: Missing values for variables out percent, average income, housework, and non-farm income.

In the above equation, $Status1_i$ and $Status2_i$ are the dependent variables, which represent the individual's status in the families. *Landout* and *Landin* are the focus variables. $\alpha_1$ and $\alpha_2$ are the intercept terms. $\beta_i, \gamma_i, \delta_i,$ and $\delta_i,$ in front of the explanatory variables and control variables, are the parameters to be estimated ($i$ = 1, 2, 3, 4). The above parameters were set to test the impact of agricultural land transfer on the status of women in rural households in the household. $X_1$, $X_2$, and $X_3$ are the factors affecting the status of women in rural households at the individual, household, and village levels, respectively. $\varepsilon_i$ is a random disturbance term. Considering that *Status*1 is an ordered discrete variable, ordered logit regression is mainly used to improve the fitting effect. *Status*2 is a continuous variable, so OLS regression is used properly.

## 5. Empirical Results

### 5.1. Baseline Regression

Table 5 presents the estimated results of the impact of rural land transfer on the status of women in households. After controlling for individual, household, and village level characteristics, there is a positive impact of rural land transfer on the status of women in households. The result is significant at the 5% confidence level, meaning that the status of women is higher in rural land transfer-out households. Although the effect of rural land transfer-in on the status of women in the family is not statistically significant, it still shows a negative effect, to some extent, which indicates that rural land transfer is not conducive to improving the status of women in rural households. Ordered Logit and OLS regression results are consistent, with the estimated coefficients of 0.1537 and 0.0734 for rural transfer-out, respectively, both of which are significantly positive at the 5% level when all else is held constant. The OLS results suggest that the status of women in households with farmland transfer-out is 0.0734 higher than that of women in households without farmland transfer-out. The transfer-out of farmland improves the status of women in households, while the transfer-in of farmland does not. In addition to land rent, women transferring out of rural land can earn much higher independent income through off-farm employment than

in agricultural production. However, most women in households that have transferred to agricultural land do not have access to independent income. The increase in private labor that is not monetarily remunerated. The weak access to resources is not conducive to the improvement of the status in the household. In summary, hypothesis H1 is tested.

**Table 5.** Impact of rural land transfer on the status of women in rural households for all rural female samples.

| Variables | Model (1) | Model (2) | Model (3) | Model (4) |
|---|---|---|---|---|
| *Landout* | 0.1537 ** −0.0749 | | 0.0734 ** −0.0366 | |
| *Landin* | | −0.0593 −0.0623 | | −0.0242 −0.0293 |
| Individual-level | YES | YES | YES | YES |
| Household-level | YES | YES | YES | YES |
| Village-level | YES | YES | YES | YES |
| Province-fixed effect | YES | YES | YES | YES |
| Adj (Pseudo) $R^2$ | 0.0488 | 0.0487 | 0.1295 | 0.1291 |
| Obs | 6975 | 6975 | 6975 | 6975 |

Note: ** denotes passing 5% significance tests, respectively, and the values in square brackets below the coefficients are robust standard errors;The total number of rural female samples participating in the regression is 6975, due to the presence of missing values for variables out percent and average income.

### 5.2. Robustness Test

In the above baseline regression, there is a situation in the sample in which the behavior of transferring in and out of farmland occurs at the same time. This behavior may affect the true accuracy of the estimation results. Drawing on the practice of Yang and Deng et al. [55], 89 females in the sample whose families have both farmland transfer-in and farmland transfer-out behaviors are excluded from the sub-sample regression. The results are shown in Table 6, where the direction of the impact of farmland transfer-out and farmland transfer-in on the status of women in households remains the same. The estimated coefficients of farmland transfer-out and farmland transfer-in are slightly improved, showing the robustness of the baseline regression results.

**Table 6.** Robustness tests: Subsample regression for all rural female samples.

| Variables | Model (1) | Model (2) | Model (3) | Model (4) |
|---|---|---|---|---|
| *Landout* | 0.1676 ** −0.0788 | | 0.0782 ** −0.0385 | |
| *Landin* | | −0.06 −0.0644 | | −0.0255 −0.0303 |
| Individual level | YES | YES | YES | YES |
| Household-level | YES | YES | YES | YES |
| Village level | YES | YES | YES | YES |
| Province-fixed effect | YES | YES | YES | YES |
| Adj (Pseudo) $R^2$ | 0.049 | 0.0488 | 0.1297 | 0.1292 |
| Obs | 6891 | 6891 | 6891 | 6891 |

Note: ** denotes passing 5% significance tests, respectively, and the values in square brackets below the coefficients are robust standard errors; The sample before exclusion included 6975 rural female respondents, due to the presence of missing values for variables out percent and average income.

### 5.3. Endogeneity Discussion

5.3.1. Selectivity Bias

To alleviate the endogeneity problem caused by the possible selectivity bias of the sample, this paper utilizes the propensity to match score method (PSM). In this research, farmers who have transferred their agricultural land are set as the treatment group and farmers who have not carried out land transfer are set as the control group. The average treatment effect (ATT)[1] of transferring out and transferring in agricultural land is estimated

by using nearest neighbor matching (k-value of 4), caliper matching (caliper is set to 0.01), and kernel matching (quadratic kernel with a bandwidth of 0.01). The results in Table 7 are as follows. The ATT obtained from near-neighbor matching, caliper matching, and kernel matching, provide further evidence that farmland transfer-out improves the status of women in rural households. Taking the results of near-neighbor matching as an example, the status of women in households of farmland transfer-out households improves by 10.3%. The results of caliper matching and kernel matching show that the status of women in households can be improved by 7.7%. The effect is not obvious in the case of farmland transfer-in, which does not pass the significance test. The results of the Propensity Matching Score (PSM) method also demonstrate that the baseline regression is robust.

**Table 7.** Endogenous treatment: propensity score matching for all rural female samples.

| Variables | Matching Method | ATT (*Landout*) | t-Value | ATT (*Landin*) | t-Value |
|---|---|---|---|---|---|
| | neighbor matching | 0.1026 ** | 2.52 | −0.0217 | −0.66 |
| *Status2*[2] | caliper matching | 0.0765 ** | 1.98 | −0.0239 | −0.77 |
| | kernel matching | 0.0768 ** | 1.99 | −0.0249 | −0.8 |

Note: The total number of rural female samples participating in the regression is 6975, due to the presence of missing values for variables out percent and average income; ** denotes passing 5% significance tests, respectively, and the values in square brackets below the coefficients are robust standard errors.

5.3.2. Omitted Variable Bias

To minimize the inaccuracy of the estimation results brought by omitted variable bias, referring to the research of Guo and Ma [56], this paper needs to discriminate the magnitude of the bias intensity, brought by unobservable variables by observable variables. In short, this article proposes to construct the Ratio index using regression with three differentiated control sets:

$$\text{Ratio} = \left| \frac{\hat{\beta}_2}{\hat{\beta}_2 - \hat{\beta}_1} \right| \tag{5}$$

In Equation (5), $\hat{\beta}_2$ is the estimated coefficient of the focus variable after controlling for all observable variables and $\hat{\beta}_1$ is the coefficient of the focus variable after controlling for limited observable variables. When the Ratio is larger, the explanatory power of the control variables incorporated within the selected model is stronger. The model can be considered less likely to have omitted variable bias. If Ratio > 1, the omitted variables do not have strong explanatory power for the estimation results, compared to the control variables already in the model. The omission bias at this point is negligible. Table 8 shows the omitted variable bias test for the baseline regression in this paper. In the Ratio test, we construct three pools separately: the first pool incorporates only the focus variables; the second pool incorporates individual and household control variables; and the third pool incorporates all control variables. As shown in the table below, pools 1–2, 2–3, and 1–3 compute Ratio values of 15.5, 4.19, and 3.47, respectively, all of which are greater than 1. If there are unobserved variables that lead to omitted-variable bias, they are required to have explanatory power of at least 3.47 times that of the controlled variables. From the results of the test, even the presence of unobserved variables is not enough to bias the estimates after controlling for variables at the individual, household, and village levels.

In empirical studies, the main sources of endogeneity problems include selectivity bias, omitted variables, and reverse causality. Propensity score matching (PSM) is considered to be an effective method for addressing selectivity bias, and this paper is no exception for selectivity bias. For omitted variables, this article will test whether omitted variable bias affects the empirical results by constructing Ratio indices from regressions of three differentiated control sets. From intuition and previous studies, there is no obvious reverse causality between rural land transfer and the status of women in households, and thus it is not addressed. As a result, somehow the endogeneity problem of this paper is controlled.

**Table 8.** Endogeneity treatment: Ratio value test for all rural female samples.

| Dependent Variable | Coefficients | Ratio Value |
|---|---|---|
| Pool 1 (No control variables) | 0.0522 | Pool 1–2: 15.50 |
| Pool 2 (individuals and households) | 0.0558 | Pool 2–3: 4.19 |
| Pool 3 (individuals, households and villages) | 0.0734 | Pool 1–3: 3.47 |

Note: The total number of rural female samples participating in the regression is 6975, due to the presence of missing values for variables out percent and average income.

*5.4. Heterogeneity Analysis*

In 2004, China established 13 provinces as major grain-producing areas to guarantee the security of the national grain supply. The main grain-producing areas have a higher degree of land transfer due to their topographical features, policy support, and other factors. The corresponding proportion of those who help others to do farm work or go out to work should be higher in the major grain-producing areas. However, the statistics of the sample data show that the proportion of helping others to do farm work or going out to work in major grain-producing areas is around 60%. The same applies to nonmajor grain-producing areas. The results in Table 9 show that the estimated results of the impact of farmland transfer on the status of women in rural households in nonmajor grain-producing areas are significant. Furthermore, the estimated coefficient (0.1170) is larger than that in major grain-producing areas. So, what causes this discrepancy? The reason may lie in the fact that, due to the limitation of cropland resource endowment in nonmajor grain-producing areas, women lack the opportunity to help others do agricultural work, so they are more inclined to go out to work after transferring out of the land. Consequently, their ability to obtain resources is improved and their status in families is enhanced. The proportion of women working outside in nonmajor grain-producing areas is larger than that in food-producing areas. Sample statistics also show that the proportion of migrant workers in the major grain-producing areas is 26%, while the proportion of migrant workers in the nonmajor grain-producing areas is 56%[3].

**Table 9.** Results of heterogeneity analysis for all rural female samples.

| Variables | Nonmajor Grain-Producing Areas | | Main Grain-Producing Areas | |
|---|---|---|---|---|
| | Model (3) | Model (4) | Model (3) | Model (4) |
| *Landout* | 0.1170 ** | | 0.0297 | |
| | −0.0526 | | (0.0503) | |
| *Landin* | | −0.3305 | | −0.063 |
| | | (0.0901) | | (0.0412) |
| Individual-level | YES | | YES | |
| Household-level | YES | | YES | |
| Village-level | YES | | YES | |
| $R^2$ | 0.1432 | 0.1419 | 0.1071 | 0.1076 |
| Obs | 3506 | 3506 | 3469 | 3469 |

Note: ** denotes passing 5% significance tests, respectively, and the values in square brackets below the coefficients are robust standard errors.

## 6. Analysis of Mechanisms

The previous paper confirms hypothesis H1 through the robustness test and endogeneity treatment that rural transfer-out can improve the status of women in rural households while farmland transfer-in cannot. In addition to this, this paper would like to further investigate through what mechanism the effect of farmland transfer on the status of women in rural households transmitted. We will move forward to discuss the transmission mechanism below.

### 6.1. Mechanism Variables and Modeling

Rural land transfer affects women's status by influencing their resource-grabbing capacity. Established studies have identified higher female income as an important channel for the status in household improvement [7,21]. Ding et al. [9] and Li et al. [7] argued that female household division of labor determines women's status in households. The degree of participation in daily affairs in the household affects women's status in the family. Thus, the time spent on household chores can laterally corroborate women's resource-grabbing ability. Combined with the theoretical mechanism specifically, the less time women spend on household chores, the more time they spend on non-agricultural labor, and the more income they receive accordingly, the higher their status in the family. Based on the availability of data, this paper chooses "time spent on household chores" and "income from labor" as mechanism variables to represent women's resource acquisition ability. In order to test this mechanism, this paper draws on the analysis method of Jiang [57] on the mediating effect and sets up the model as follows:

$$M_i = \alpha_3 + \beta_5 \times Landout + \gamma_5 \times X_1 + \delta_5 \times X_2 + \theta_5 \times X_3 + \varepsilon_5 \tag{6}$$

$$M_i = \alpha_4 + \beta_6 \times Landin + \gamma_6 \times X_1 + \delta_6 \times X_2 + \theta_6 \times X_3 + \varepsilon_6 \tag{7}$$

$M_i$ is an individual's income from labor, called "non-farm income", and time spent on household chores, called "housework", as mechanism variables, and others are the same as in the baseline regression.

### 6.2. Results of the Analysis of Mechanisms

As shown in Table 10, transferring out of rural land significantly increases female non-farm income and reduces the time spent on housework while transferring in of agricultural land has a negative but not significant effect on female income impact and time spent on housework. Specifically, after controlling for the relevant variables, the non-farm income of households with transferred out farmland increased by 5859.386 yuan, compared to the income of females who have not transferred out rural land. The result is significant at the 1% level; there is a significant negative impact of farmland transfer-out on female participation in housework time, with an estimated coefficient of −0.1579. The estimation result is significant at the 5% level. It means that rural land transfer-out reduces the time spent by females participating in household chores by about 0.158 h per day. The regression results indicate that the resource-grabbing capacity of women is improved by engaging in non-farm labor and reducing the time spent on housework after the transfer of agricultural land. Rural land transfer-out promotes the redistribution of housework, which can significantly reduce women's burden of household work and increase their participation in non-farm labor, providing more time and opportunities for women to realize their self-worth, create economic value, and improve their status in families. The impact of rural land transfer-in on women's income is negative but not significant, for this paper tries to give a possible explanation: women are usually not paid directly in monetary terms for their domestic work. Rural land transfer-in strengthens women's bondage to land and family and creates the opportunity cost for not being able to engage in non-farm labor to earn an income. As a result, both of these further undermine women's ability to capture resources. In addition, for women who can migrate to the big cities, entering the big cities does not only mean an increase in income levels but also a change in gender attitudes. The aggregation and interaction effects of migrant women help to build a modern sense of femininity and demand for rights and to improve their status in families.

**Table 10.** Mechanism analysis results for all rural female samples.

| Variables | Model (6) | | Model (7) | |
|---|---|---|---|---|
| | **Mechanism Variables** | | | |
| | **Non-Farm Income** | **Housework** | **Non-Farm Income** | **Housework** |
| *Landout* | 5859.386 *** (1182.016) | −0.1579 ** (0.0687) | | |
| *Landin* | | | −697.9678 (702.937) | −0.0168 (0.0632) |
| Individual level | YES | | YES | |
| Household-level | YES | | YES | |
| Village level | YES | | YES | |
| Province-fixed effect | YES | | YES | |
| Adj (Pseudo) $R^2$ | 0.0598 | 0.0512 | 0.0543 | 0.0506 |
| Obs | 6975 | | 6975 | |

Note: The total number of rural female samples participating in the regression is 6975, due to the presence of missing values for variables out percent and average income; ** and *** denote passing 5% and 1% significance tests, respectively, and the values in square brackets below the coefficients are robust standard errors.

Considering that there may be a bidirectional causal relationship between rural land transfer and non-farm income and that non-farm income may affect the transfer of agricultural land, this paper tries to find an instrumental variable to solve the endogeneity problem in the mechanism test. We draw on the research method of Liu et al. [58] to select "village land flatness" as an instrumental variable. The reasons for "village land flatness" as an instrumental variable are as follows: on the one hand, the village landscape affects the difficulty of land transfer to a certain extent, the flatter the land is, the more difficult it is to transfer, and the easier the land is to transfer. On the one hand, the topography of the village affects the difficulty of land transfer to a certain extent. The flatter the land, the easier it is for land transfer. It meets the requirement of instrumental variable correlation. On the other hand, as an exogenous variable, the topography of the village does not directly affect women's income from work, so it meets the requirement of homogeneity of the instrumental variable.

From the regression results in Table 11, the first stage instrumental variable has a positive effect on rural land transfer-out at a 10% significance level. The first stage F-value of 11.31 is greater than the empirical value of 10 and is greater than the critical value of error bias of 8.96 as specified by Stock-Yogo [59]. The estimation results show that there is no problem of weak instrumental variables. The results of the second stage model regression suggest that rural land transfer-out has a significant positive effect on non-farm income and the estimated coefficient of 40,690.67 is significantly higher than 5859.386 before the use of instrumental variables. The significance is reduced to 10%, which indicates that the endogeneity problem of rural land transfer-out is mitigated. We can conclude that rural land transfer-out significantly improves women's income from labor. Rural land transfer-out improves women's status in the household by increasing women's non-farm participation and non-farm income. In summary, hypothesis H2 is tested while hypothesis H3 is not tested.

**Table 11.** Endogeneity test of the impact of farmland transfer on non-farm income.

| Variables | Non-Farm Income | |
|---|---|---|
| | **OLS** | |
| *Landout* | 0.0013 *** (0.0004) | |
| IV: village land flatness | | 40,690.67 * (22,823.65) |
| Individual-level | YES | YES |
| Household-level | YES | YES |

**Table 11.** *Cont.*

| Variables | Non-Farm Income | |
| --- | --- | --- |
| | OLS | |
| Village-level | YES | YES |
| Province-fixed effect | YES | YES |
| F-value | 11.59 | |
| Obs | 6676 | 6676 |

Note: The total number of rural female samples participating in the regression is 6676, due to the presence of missing values for variables housework and non-farm income; * and *** denote passing 10% and 1% significance tests, respectively, and the values in square brackets below the coefficients are robust standard errors.

## 7. Conclusions and Recommendations

The study presents several key findings. Firstly, rural land transfer-out significantly enhances women's household status, while the impact of rural land transfer-in on women's household status remains insignificant. Subsample estimation and the Propensity Matching Score Method (PSM) were employed for additional testing, confirming the robustness of the estimation results. Secondly, heterogeneity analysis reveals that rural land transfer-out in non-major grain-producing areas has a more pronounced positive effect on female household status compared to major grain-producing areas. The reason for this is that women in non-food-producing regions have more possibilities for non-farm employment. The mechanism test further validates the transmission pathway of the impact of rural land transfer on female household status. The use of the Two-Stage Least Squares (2SLS) treatment helps mitigate the potential bidirectional causality between rural land transfer-out and out-of-home labor. Thirdly, rural land transfer-out emerges as a significant driver for improving the status of women in rural households through the redistribution of household labor and the increase in off-farm independent income. Conversely, rural land transfer-in does not contribute to enhancing women's independent income, thereby failing to promote women's status in rural families. The transfer-out of agricultural land increased women's income by approximately $5859.386 and reduced the amount of time spent on housework by 0.158 h per day. The conclusion of this paper is a new understanding for us to explore the change of rural women's status in the family. The transfer of agricultural land plays a certain role in the improvement of women's family status.

The study's findings lead to the following recommendations:

1. **Accelerate Rural Land Transfer, Liberate Rural Women and Promote Non-agricultural Employment:** According to the research conclusion, women who have transferred in rural land, engage in low-income agricultural labor and lack monetary income, which leads to economic dependence on their husbands [33]. The employment quality and monetary income of women who transfer in rural land are lower than those of rural women who transfer out rural land, which is not conducive to the improvement of their own status [48]. We should encourage women to transfer small plots of land to new types of management and develop land trusteeship services to liberate rural women. Economic empowerment can improve women's status at home [43]. Compared with agricultural labor, rural women engaged in non-agricultural labor can improve their ability to capture resources and empower themselves.

2. **Increase Women's Public Labor Participation and Paid Labor Time:** Women who transferred out the land have 0.158 h less time per day for household chores, and accordingly have increased paid labor and monetary income and gained economic empowerment. As mentioned in the theoretical analysis, the redistribution of housework after transferring out the land can ensure that women receive more economic income based on their service contribution to the family [25]. The status in the family can be improved with the increase of paid labor. We advocate the sharing of household chores between husband and wife and the socialization of household chores, so that more rural women can get rid of the constraints of "raising children with their

husbands and doing the laundry and cooking", get out of the family and go to the workplace, realize their own economic value and improve their status in households.

3. **The Problem of Rural Women's Employment After Rural land Transfer-out is A Matter of Concern:** The impact of the transfer of agricultural land on women's status in the family is heterogeneous between the major grain-producing areas and the nonmajor grain-producing areas. After transferring out their land, women in the major grain-producing areas lacked possibilities for non-farm employment compared with women in the nonmajor grain-producing areas. The stigmatization of women working outside the home also discourages women from engaging in non-farm employment [33]. Non-monetary contributions to the household by women who stay at home do not improve their status in the family and even reduce income [60]. Broadening the sources of information and income, promoting rational employment for rural women left behind, and providing training and employment opportunities close to their homes for low-income and low-education women in the major grain-producing areas can improve their status in rural households.

The status of women within families is a topic of shared interest in the fields of economics and sociology. This paper contributes to this discourse by examining the issue through the lens of rural land transfer. Initially, the lower status of women in rural households has been a prevalent stereotype, resulting in a relative lack of scholarly attention. This study aims to challenge and alter this stereotype by investigating the impact of rural land transfer on the status of women in rural families, thereby making a modest contribution to the broader understanding of women's status in rural settings. Furthermore, existing research has commonly asserted that rural women face a deficit in resources. However, this paper contends that land can indeed serve as a valuable resource for rural women. Proper utilization of land has the potential to empower rural women and enhance their family status. Drawing inspiration from Engels's women's theory, this paper traces the evolution of resource determination theory, establishing a novel analytical framework for understanding the status of women in rural households.

Despite these contributions, the paper acknowledges certain limitations stemming from data constraints. Firstly, the analysis does not delve into the long-term effects of rural land transfer on women's family status over an extended period. Additionally, the use of a dummy variable for rural land transfer prevents a more detailed exploration of its heterogeneous impact on women's status concerning the scale of land transfer. Finally, religious affiliation may affect women's status, but we cannot explore this further due to data limitations. Subsequent data addressing these limitations would significantly enhance the meaningfulness of this study.

**Author Contributions:** Conceptualization, M.H. and D.Z.; methodology, D.Z.; software, D.Z.; validation, M.H., D.Z. and L.L.; formal analysis, D.Z.; investigation, D.Z.; resources, D.Z.; data curation, D.Z.; writing—original draft preparation, D.Z.; writing—review and editing, M.H. All authors have read and agreed to the published version of the manuscript.

**Funding:** This study is funded by the National Natural Science Foundation of China project "Research on Spatial Distance, Relationship Strength, and Contract Performance Mechanism of Agricultural Land Transfer", with approval number 72163003.

**Data Availability Statement:** The associated data set in the study are available upon request.

**Conflicts of Interest:** The authors declare no conflicts of interest.

## Notes

1. Most of the observations matched by the three matching methods are within the common range of values, and the quality of the matching is reliable, satisfying the Common Support Condition (CSC).

2. *Status*2 and *Status*1 estimation results are the same, OLS estimation results are easy to explain, so choose here to choose *Status*2 as an explanatory variable, the same as below.)

[3]　　The proportion of out-of-home labor cannot be measured directly due to data limitations, and in this paper, the income from out-of-home labor of a farm household is defined as 0 as staying at home, otherwise, it is out-of-home labor.

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
