# Peer review of "Study of the Impact of Rural Land Transfer on the Status of Women in Rural Households"

_land, doi:10.3390/land13010107_

Round 1
Reviewer 1 Report
Comments and Suggestions for Authors
I would like to thank for the opportunity to read this manuscript. Congratulations to the authors - the article is original, interesting, and topical!
The article is of good publishing quality, but there is still space to add a bit to its readability:
First, it is recommended to use the full working for the acronym 'OLS' in the Abstract.
Second, in lines 9-10, 54-56, and 214-217 it is stated, that in this study authors use data from the 2014 China Family Panel Studies (CFPS2014), which is "a nationwide 215 tracking survey program conducted every two years by Peking University's Social Science 216 Research" (lines 215-217). There should be given a clear explanation, of why the almost 10-year-old data is used for this research. Why the more recent data is not used from this survey, carried out every two years? There is a short hint, that simply "only the 2014 questionnaire contains information on respondents' family status and farmland transfer..."(lines 222-223), but there should be better explained in the article how specifically this kind of data from the year 2014 help in answering the main research questions (lines 51-54).
Finally, in "Conclusions and insights", it is suggested to remove the first three sentences (lines 517-521), which actually have nothing to do with conclusions; it only repeats the text of the manuscript concerning the used data and applied methods and tools for analysis. This has already been stated in the Abstract and Chapter 4.
With best regards,
The reviewer
Author Response
Dear reviewer:
I hope this email finds you well. Thank you for your prompt and thorough review of our manuscript titled”[Study of the impact of rural land transfer on women’s family status]” (Manuscript ID: land-2746445).
We appreciate the time and effort you have dedicated to evaluating our work. Your insightful comments and constructive feedback have been invaluable in enhancing the quality of our manuscript.
In response to your comments, we have carefully revised the manuscript, addressing each of the concerns and incorporating the suggested improvements. Below we will reply to your review comments one by one. A detailed summary of the changes made is provided in the attached document.Please check it when you have free time
Once again, we appreciate your time and expertise in reviewing our manuscript and thank you for your guidance on my academic writing path. We look forward to receiving your final feedback and hope that the revised version meets the standards of publication in Land.
With best regards,
Corresponding Author: Donglai, Zhou

Reviewer 2 Report
Comments and Suggestions for Authors
Author Response
Dear reviewer:
I hope this email finds you well. Thank you for your prompt and thorough review of our manuscript titled”[Study of the impact of rural land transfer on women’s family status]” (Manuscript ID: land-2746445).
We appreciate the time and effort you have dedicated to evaluating our work. Your insightful comments and constructive feedback have been invaluable in enhancing the quality of our manuscript.
In response to your comments, we have carefully revised the manuscript, addressing each of the concerns and incorporating the suggested improvements. The specific modifications are in the attachment.Please check it when you have free time.
If you have any further questions or require additional information, please do not hesitate to contact us. We are more than willing to provide any clarification or address any remaining concerns you may have.
Once again, we appreciate your time and expertise in reviewing our manuscript and thank you for your guidance on my academic writing path!. We look forward to receiving your final feedback and hope that the revised version meets the standards of publication in Land.
we wish you success in your work and a good mood!
With best regards,
Corresponding Author: Donglai, Zhou

Reviewer 3 Report
Comments and Suggestions for Authors
1. The introduction and theoretical framework are quite interesting, however, why only quote Marx and Engels? what about contemporary theories or researcher that discuss woman´s labour and feminization of the rural areas? I strongly suggest to update it because a lot of things changed in the last 200 years in rural areas.
2. What about rural culture? I understand that this study is focused on data analysys, but seems rural rights transfers are more complicated that the data show. What about womans beliefs? did every place you survey is the same? please give a little bit of context for the people that is not Chinese.
3. Add some images of the field work!
4. I don´t thinks this chapter provides "a new insight" this should be a policy paper that will help for decision making.
5. The conclusions are quite debatable, please provide recommendations. "Based on the findings of the study, we come up with the following insights: Firstly, 538
accelerating the transfer of agricultural land and promoting moderate-scale management 539
will liberate rural women's labor. Encourage the transfer of small plots of land to new 540
management bodies, develop land trusteeship services or land management on an appro- 541
priate scale, and promote agricultural mechanization to liberate the rural labor force, es- 542
pecially the rural women's labor force. Secondly, increase women's public labor hours. On 543
the one hand, inclusive childcare and custodial services should be developed to enrich the 544
after-school cultural life of rural children and to share the time costs of childcare and child- 545
care for rural women. On the other hand, we are advocating the sharing of household 546
chores between husband and wife and the socialization of household chores, so that more 547
rural women can get rid of the constraints of "teaching their children and doing the laun- 548
dry and cooking" and go into the workplace to realize their economic value and improve 549
the status of their families, which will help to improve the outlook on family reproduction. 550
Thirdly, we are concerned about the employment of rural women after the transfer of 551
agricultural land. The transfer of land has polarized rural women's employment: young 552
women go out to work while middle-aged and old women face unemployment. It is im- 553
portant to broaden sources of information and income, promote reasonable employment 554
for women left behind in rural areas, and provide training and employment opportunities 555
for low-income and low-education women close to their homes so that the benefits of the 556
transfer of agricultural land can spill over. "
Author Response
Dear reviewer:
I hope this email finds you well. Thank you for your prompt and thorough review of our manuscript titled”[Study of the impact of rural land transfer on women’s family status]” (Manuscript ID: land-2746445).
We appreciate the time and effort you have dedicated to evaluating our work. Your insightful comments and constructive feedback have been invaluable in enhancing the quality of our manuscript.
In response to your comments, we have carefully revised the manuscript, addressing each of the concerns and incorporating the suggested improvements.Detailed modifications can be viewed in the attachment.Hope you can check it out when you have free time.
If you have any further questions or require additional information, please do not hesitate to contact us. We are more than willing to provide any clarification or address any remaining concerns you may have.
Once again, we appreciate your time and expertise in reviewing our manuscript and thank you for your guidance on my academic writing path!. We look forward to receiving your final feedback and hope that the revised version meets the standards of publication in Land.
we wish you success in your work and a good mood!
With best regards,
Corresponding Author: Donglai, Zhou

Round 2
Reviewer 3 Report
Comments and Suggestions for Authors
Many thanks for the corrections of the paper. It improved in several areas, however, I insist in:
1. The literature review is still poor, it is good that you search for contemporary scholars, however literature review should not end in quoting Lenin as a reference (not because he is not important, because you are working on current situation in rural China).
2. I suggest that literature review should be a theoretical framework because as a "review" it is too brief.
3. Maybe you can add a table between different periods to understand how the situation for woman in China is changing. I mean, a period table where you quote Marx and Engels, then a 20th Century scholars review and then 21st Century with current research.
4. I think you are not been enough critical about patriarcal context, you state that is can be cultural and use for different purposes. I did not like it at all.
5. If you are working on gender, specially on rural woman, you should be more critical, specially in your recommendations. I strongly suggest that not only your data is used to support your findings, also the scholars and researchers that you quote in your "literature review".
6. Your paper has very interesting data findings, but it get lost among variables to realy discuss of the situation of rural woman in China (with the rural transfers).
7. Your recommendations and conclusions must be improved.
Author Response
Dear reviewer, I'm sorry to bother you on your vacation. We want you to know that we value your opinion very much. Your opinion is very important to improve the quality of our paper.We completely revised the manuscript in accordance with your comments. The modification Instructions is attached, please check it when you have free time.Please see the attachment. Again, I apologize for disturbing you over the holidays and wish you a good mood! Wish you a happy New Year in advance!Wish you all the best.
Donglai Zhou Author2 & Corresponding author Email: xd_dlzhou@163.com
Round 3
Reviewer 3 Report
Comments and Suggestions for Authors
Many thanks for the corrections and added information. Just check your Abstract that match the new version or your paper.
Author Response
Dear reviewer:
I hope this email finds you well. Thank you for your prompt and thorough review of our manuscript titled”[Study of the impact of rural land transfer on women’s family status]” (Manuscript ID: land-2746445).
We appreciate the time and effort you have dedicated to evaluating our work. Your insightful comments and constructive feedback have been invaluable in enhancing the quality of our manuscript.
In response to your comments, we have carefully revised the manuscript, addressing each of the concerns and incorporating the suggested improvements.
In the last revision, we focused on revising the theoretical framework, heterogeneity analysis and conclusions in accordance with your valuable comments. Therefore, we make new additions to the theoretical framework, heterogeneity analysis and conclusion in the abstract, in order to make the content of the paper match the abstract.Please see the attachment.
If you have any further questions or require additional information, please do not hesitate to contact us. We are more than willing to provide any clarification or address any remaining concerns you may have.
Once again, we appreciate your time and expertise in reviewing our manuscript and thank you for your guidance on my academic writing path!. We look forward to receiving your final feedback and hope that the revised version meets the standards of publication in Land.
we wish you success in your work and a good mood!
With best regards,
Corresponding Author: Donglai, Zhou
